# Bubble Properties in Bubbling and Turbulent Fluidized Beds for Particles of Geldart's Group B

**Tom Wytrwat [1],*** [ID]**, Mahdi Yazdanpanah [2] and Stefan Heinrich [1]** [ID]

[1]    Institute of Solids Process Engineering and Particle Technology, Hamburg University of Technology, Denickestrasse 15, 21073 Hamburg, Germany; stefan.heinrich@tuhh.de

[2]    Total Research & Technology Gonfreville (TRTG), 76700 Harfleur, France; mahdi.yazdanpanah@total.com

*    Correspondence: tom.wytrwat@tuhh.de

**Abstract:** Predicting bubble properties in fluidized beds is of high interest for reactor design and modeling. While bubble sizes and velocities for low velocity bubbling fluidized beds have been examined in several studies, there have been only few studies about bubble behavior at superficial gas velocities up into the turbulent regime. For this reason, we performed a thorough investigation of the size, shape and velocity of bubbles at superficial gas velocities ranging from 0.18 m/s up to 1.6 m/s. Capacitance probes were used for the determination of the bubble properties in three different fluidized bed facilities sized of 0.1 m, 0.4 m and 1 m in diameter. Particles belonging to Geldart's group B (Sauter mean diameter: 188 μm, solid density: $\rho_s$ = 2600 kg/m$^3$) were used. Correlations for the determination of bubble phase holdup, vertical bubble length and bubble velocity are introduced in this work. The shape of bubbles was found to depend on superficial gas velocity. This implies that at large superficial gas velocities the horizontal size of a bubble must be much smaller in comparison to its vertical size. This leads to a decrease of pressure fluctuations, which is observed in the literature as a characteristic of transitioning into a turbulent regime.

**Keywords:** bubbling; turbulent; fluidized bed; regime transition; Geldart group B; capacitance probe; phase holdup; bubble size; bubble velocity

---

## 1. Introduction

Bubble properties in a fluidized bed are of high interest for reaction modeling and the prediction of the behavior in scale-up. If the size, shape, distribution and velocity of bubbles are known, the volume-specific area of the bubble phase can be predicted, which is an important factor for modeling mass-transfer behavior. Furthermore, it enables predicting the expansion of the fluidized bed and consequent impact on the reactor design and reaction conversion.

Most investigations of bubble properties in fluidized beds were done in the bubbling fluidized bed regime [1]. Different methods of measuring were used, such as surface photography, photography in quasi two-dimensional fluidized beds, capacitance probes, fiber optical probes, electro-resistive probes, X-ray photography, magnetic resonance imaging and more [1,2]. Capacitance probes were used by several authors as a reliable tool for fluid-dynamic and bubble flow investigations [3–9]. This type of probe can be operated in extreme conditions, such as high temperature beds, for both the dilute and dense phases with high solids concentrations [10–12].

Capacitance probes measure temporal local changes in the solid concentration. The simultaneous measurement of two channels aligned vertically above each other allows to determine the time lag of sudden changes in concentration induced by bubbles rising in the fluidized bed [3]. Because movements of bubbles are much larger in the vertical direction than horizontally, the local measurements of these concentration variations give information about vertical movements and bubble sizes only.

Bubbles in fluidized beds are three dimensional and their horizontal size can only be determined by imaging measurement techniques, such as magnetic resonance imaging of miniature size plants or photography in quasi two-dimensional fluidized beds. The shape of bubbles in fluidized beds is known to be non-spherical. In the bubbling fluidized bed flow regime several studies were conducted to determine bubble shapes slightly above the minimum fluidization velocity [1]. Most of the authors described bubbles as spherical caps pushing a cloud of particles above them and dragging particles in a wake below them [1,4]. The shape of a bubble was found to be of a cap, with different sizes in the horizontal and vertical directions. Determining the vertical length of a bubble using capacitance probes poses a challenge: depending on the radial position of the probe in which it penetrates in the bubble, different values will be measured. A stochastic distribution is thus observed [13], and instead of referring to a definite vertical length, the term "pierced length" is employed. Superimposed to this are the statistic processes of bubble formation, movement, coalescence and preferred flow paths, leading to local distributions of bubble sizes [4]. For this reason, vertical sizes measured are a product of the overlapping distributions of size and pierced length.

Horizontal bubble sizes can only be estimated from capacitance probe measurements when the ratio of vertical bubble size to horizontal bubble size is known. Bar-Cohen et al. [5] assumed the vertical bubble size to be equal to the volume-equivalent diameter (bubbles are spheres), whereas Werther [7] described the bubbles based on spherical caps as rotationally symmetrical ellipsoids with a size-independent shape factor. Some authors assumed a shape factor of 1.6 to calculate the volume-equivalent diameter from the pierced length [13–15].

Most research about bubble sizes in a turbulent regime or at the regime transition from bubbling to turbulent was performed for particles of Geldart group A [16–19]. In fluidized beds with these particles, bubbles attain a maximum size and break up. Consequently, bubble size does not increase even if the gas flow rate is increased. This corresponds to a maximum in the pressure fluctuations, which is the most conventional regime change measurement technique [16]. This phenomenon results in high heat and mass transfer in a turbulent fluidized bed in comparison to a bubbling fluidized bed thanks to higher particle and gas mixing.

For beds containing particles of Geldart group B, Lee and Kim found a decrease of bubble sizes in the turbulent regime just as for smaller particles [20]. By contrast, Werther and Wein and Andreux et al. measured larger bubble sizes with increasing superficial gas velocity even beyond the point of transition to the turbulent regime under similar conditions [21,22]. These findings contradict the theory of the decrease of pressure fluctuation intensities at larger superficial gas velocities in the turbulent regime due to lower bubble sizes. The studies of Lee and Kim and Werther and Wein considered a constant shape factor of the bubbles to calculate the bubble diameter from the measured pierced length as described above [20,21]. Magnetic resonance imaging showed that bubbles are irregular in shape in the turbulent regime [23]. Thus, the assumption of the same constant shape factor for bubbles in turbulent beds as well as for bubbling beds is questionable. Bubbles in turbulent beds are generally described to be transient and to have indistinct or irregular boundaries [16]. For this reason, they are often termed void.

The behavior of bubbles in bubbling fluidized beds show similarities to the rise of large bubbles in highly viscous liquids. For the rise velocity of a single bubble ($U_{B,single}$), Davidson and Harrison [24] found the relation of the volume-equivalent bubble diameter ($d_{B,V}$) given in Equation (1), which origins from gas-liquid systems.

$$U_{B,single} \;=\; 0.711 \; \sqrt{g\, d_{B,V}} \tag{1}$$

Because bubbles in fluidized beds rise in large numbers simultaneously, they influence each other by coalescence or breakup depending on the superficial gas velocity. This swarm behavior is considered in several correlations [1]. These correlations are mostly based on the velocity of a single bubble (Equation (1) or a similar relationship of $U_B$ as a function of $d_{B,V}^{0.5}$) added by the influence of the swarm

in bubbling fluidized beds. Velocities of bubbles rising at large superficial gas velocities in turbulent beds were found to be overestimated by the correlation given by Davidson and Harrison [16,22].

In this study, superficial gas velocities were varied over a broad range from bubbling into the turbulent fluidized bed regime. In the available literature, most studies have focused on the bubbling fluidized bed regime covering only low gas velocities [1]. Only few studies exist with contradicting statements describing bubble behavior in turbulent beds of Geldart group B particles [16,19–22]. For this reason, we investigated the influence of fluidized bed size and radial and axial measurement for beds of particles of this group in this work to achieve a deeper understanding of the mechanisms on bubble properties, such as size, velocity and shape.

## 2. Materials and Methods

### 2.1. Fluidized Bed Setups

Three fluidized bed plants having diameters of 0.1 m (FB100), 0.4 m (CFB400) and 1 m (FB1000) were used for the investigation of the bubble properties. The scheme of the smallest plant is shown in Figure 1a. Superficial gas velocities in a range of 0.18 to 1.4 m/s are set with a mass flow controller (F 203AC FA by Bronkhorst High-Tech B.V.), which is supplied from a pressurized air network. A porous plate is used as the gas distributor. The fluidized bed plant is made of acrylic glass. To prevent electrostatic charging, which influences capacitance probe measurements, the inner wall of the plant is layered and grounded with a thin layer of alumina up to a height of 0.3 m. The total height of the fluidized bed is 1 m with an extended section above to prevent solids entrainment. Probe ports are installed at heights of 0.05 m (P11), 0.1 m (P12), 0.15 m (P13) and 0.2 m (P14) above the gas distributor. Measurements are carried out at two different static bed heights of 0.2 and 0.3 m. At these static bed heights, transition velocities from bubbling to turbulent fluidization of 0.98 and 1.16 m/s were measured in previous works for the same type of particles [25,26].

The scheme of the fluidized bed plant with a diameter of 0.4 m can be found in Figure 1b. Depending on the superficial gas velocity, two different roots blowers are used as air supply. In the range of 0.3 to 1 m/s the GMa 11.3 roots blower and in the range of 1 to 2 m/s the GMb 14.9 roots blower, both by Aerzen Maschinenfabrik GmbH, are used. The superficial gas velocity is measured by an orifice flowmeter. A porous plate is installed as a gas distributor. The fluidized bed has a total height of 15.6 m. Entrained solids material is separated by a cascade of two cyclones and returned into the fluidized bed via a loop-seal. Probe ports are installed at heights of 0.19 m (P21), 0.26 m (P22), 0.33 m (P23), 0.43 m (P24), 0.53 m (P25) and 0.73 m (P26). Static bed heights of 0.4 and 0.8 m are adjusted. Transition velocities from bubbling to turbulent fluidization of 0.94 and 1.09 m/s are measured at these static bed heights in previous works for the same type of particles [25,26].

The scheme of the fluidized bed plant with a diameter of 1 m is shown in Figure 1c. Analog to the plant having a diameter of D = 0.4 m, two roots blowers, the GMa 13.8 for a superficial gas velocity range of 0.3 to 0.7 m/s and the GMb 16.12 for a range of 0.75 to 1.25 m/s, both by Aerzen Maschinenfabrik GmbH, are used as an air supply. The superficial gas velocity is determined by an orifice flowmeter. The plant is equipped with a porous plate as a gas distributor. To prevent electrostatic charging the supplied air is humidified by steam. The fluidized bed has a height of 4.6 m and entrained solid material is separated in a cyclone and lead back. Probe ports are installed at heights of 0.18 m (P31), 0.53 m (P32), 0.88 m (P33) and 1.23 m (P34). The static bed height is set to 1 m. Due to a maximum superficial gas velocity of 1.25 m/s, the transition velocity from bubbling to turbulent fluidization could not be determined.

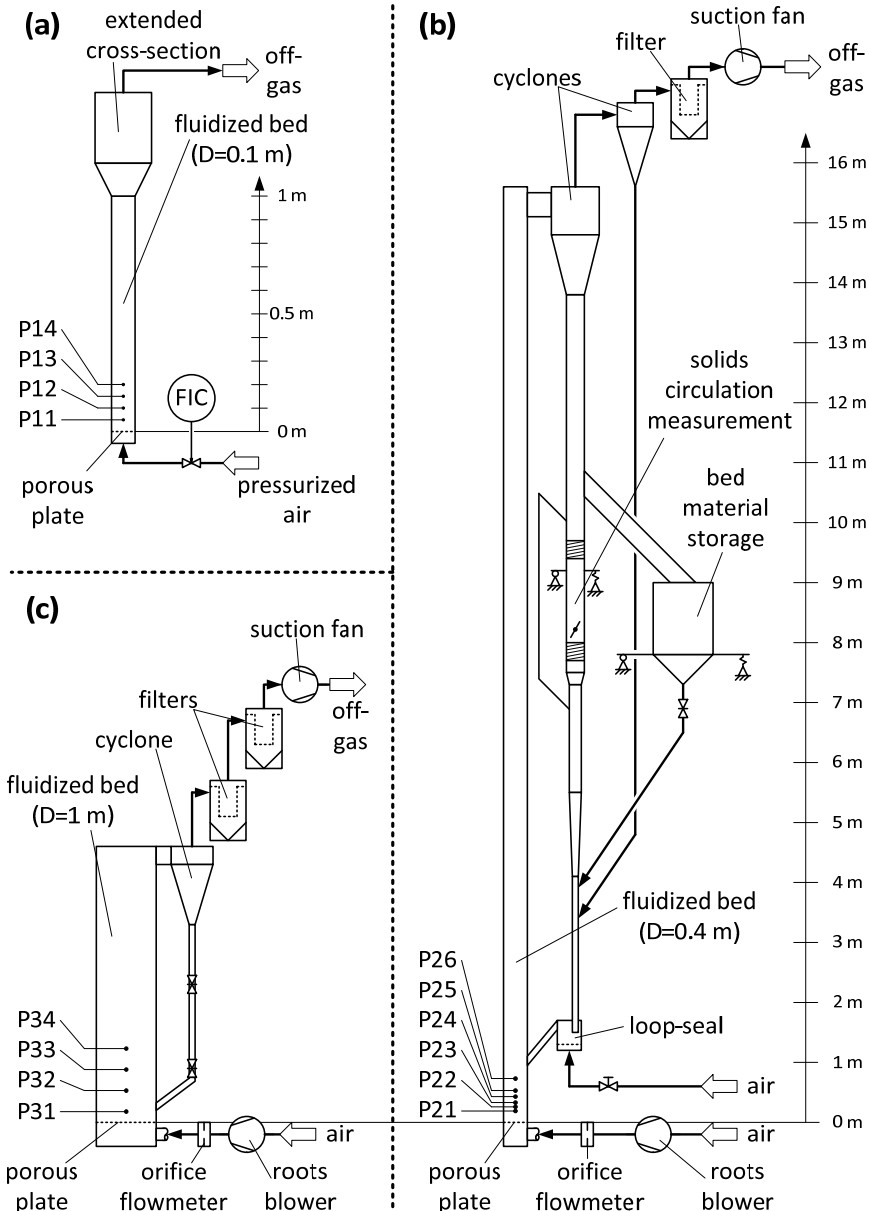

**Figure 1.** Flow sheets of the fluidized bed facilities having diameters of (**a**) $D = 0.1$ m, (**b**) $D = 0.4$ m and (**c**) $D = 1$ m including the different capacitance probe measurement ports used.

All plants are operated at ambient conditions. Quartz sand belonging to Geldart group B was used as bed material in all plants. Mean diameter $d_{50,3}$, Sauter mean diameter, solids density $\rho_s$, bulk density $\rho_b$, fixed bed solids concentration $c_{V,fb}$ and minimum fluidization velocity $U_{mf}$ of the bed material can be found in Table 1.

**Table 1.** Properties of sand particles used in this study.

| $d_{50,3}$ (μm) | SMD [1] (μm) | $\rho_s$ (kg/m³) | $\rho_b$ (kg/m³) | $c_{V,fb}$ (-) | $U_{mf}$ (m/s) |
|---|---|---|---|---|---|
| 196 | 188 | 2600 | 1405 | 0.54 | 0.073 |

[1] SMD—Sauter mean diameter.

### 2.2. Capacitance Probes

The principle of the electrical capacitance measurement technique is based on the change of the dielectric constant depending on the amount of solid entering the electric field between two electrodes. Figure 2 gives a schematic sketch of the two-channel probe tip used in this work. In this case, both channels consist of three electrodes each. In addition to the core (wolfram) and ground electrodes, a guard electrode is installed between them. This electrode shields the core from electric fields and guarantees the formation of a constant electric field for measurement. The working principle is explained in detail in [11].

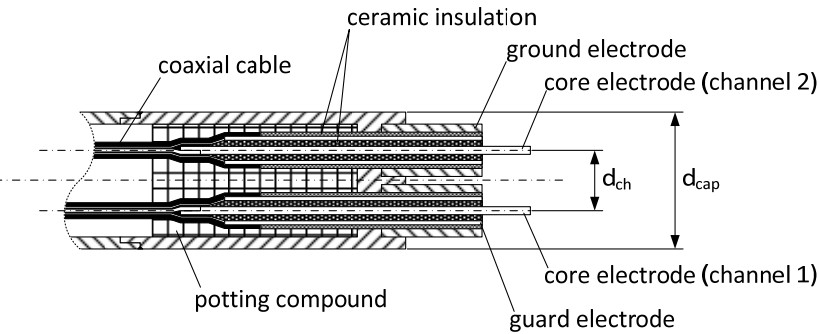

**Figure 2.** Schematic of a two-channel capacitance probe with a distance $d_{ch}$ between the channels and a total diameter of $d_{cap}$.

Three different capacitance probes were used for the investigations in the fluidized bed plants. Because capacitance probes are an invasive measurement technique, the aim was to construct probes as small as possible to minimize influences of the probe on the fluid dynamic behavior in the fluidized bed. Due to the reason of larger forces acting in larger beds in comparison to small ones, the size of the probe was constructed differently for each plant to withstand these forces. The combinations of probe diameters ($d_{cap}$), probe channel distances ($d_{ch}$) and plants, as well as probe locations, are given in Table 2.

**Table 2.** Dimensions of the different capacitance probes used.

| Probe | Used in | $d_{cap}$ (mm) | $d_{ch}$ (mm) | r/R (-) |
|---|---|---|---|---|
| CP1 | FB100 | 8 | 3.5 | 0.9, 0.8, 0.6, 0.4, 0.2, 0 |
| CP2 | CFB400 | 16 | 5.9 | 0.95, 0.9, 0.75, 0.5, 0.25, 0, −0.25, −0.5, −0.75, −0.9, −0.95 |
| CP3 | FB1000 | 22 | 6.6 | 0.96, 0.9, 0.8, 0.6, 0.4, 0.2, 0 |

The signal of each channel is treated by a pre-amplifier and amplifier (capaNCDT 600 system by Micro-Epsilon). The AD-converted voltages of the channels are recorded with a frequency of 10,000 Hz. For the determination of bubble properties, a recording duration of 10 min was chosen for each measurement point to measure an adequate amount of bubbles.

Different correlations for the calculation of the solids concentration $c_V$ from the measured voltages by capacitance probe are given in the literature [10,11]. The linear approach proposed by Hage and Werther in Equation (2) is used in this work [12].

$$c_V = c_{V,fb} \frac{U - U_f}{U_{fb} - U_f} \tag{2}$$

This approach requires knowledge about the signal levels at fixed bed concentration $U_{fb}$ and fluidizing fluid $U_f$. Both voltages are measured before and after the experiment for calibration.

For experiments using humidified air, the voltages are measured under humidified conditions. The fixed bed concentration $c_{V,fb}$ is given by the bed material used.

Because a capacitance probe is a local measurement technique, different radial positions at each axial position in the bed are measured. In the smallest plant with a diameter of 0.1 m, axial symmetry was assumed. The dimensionless radial measurement positions (r/R) are given in Table 2. Due to the high stresses acting on the probe at large depths of penetration in a plant having a diameter of 1 m, bubble properties are only measured over the radius of the plant.

### 2.3. Determination of Bubble Phase Holdup

The determination of the bubble phase holdup was done by evaluation of the local probability distribution of concentrations measured in the fluidized bed under different conditions. Examples of two histograms are shown in Figure 3. A kernel probability density distribution (non-parametric) is fitted to the concentration data, and estimates the measurement data well. Small deviations of the fit in contrast to the measured data at fixed bed concentration ($c_V = 0.54$) and air ($c_V = 0$) can be observed. The error due to these deviations was assumed to be negligible.

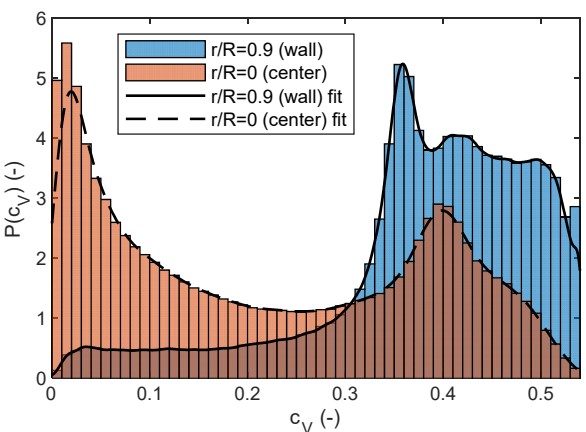

**Figure 3.** Probability density histograms at $D = 0.1$ m, $H_0 = 0.2$ m and $U_0 = 0.7$ m/s measured in the fluidized bed center (r/R = 0) and in the wall region (r/R = 0.9) with their fitted probability density distributions.

The peaks that can be found at high concentrations show the solids concentrations' distributions of the dense phase whereas the peaks at low concentrations show the occurrence of the bubble phase. Bubble phase concentrations are not always completely at a value of zero. According to two-phase theory, the bubble phase is considered empty of solids. However, bubbles contain a small particle concentration due to continuous coalescence or breakup phenomena or dispersion. The concentration depends on the size of a bubble and the location at which the probe penetrates the bubble. The capacitance probe has a fixed measurement volume. If a bubble is small or measured at the phase border both phases can be measured simultaneously, and an averaged concentration of both phases is measured in this case. The capacitance probe is an invasive measurement method. Solids can be decelerated by the probe or stuck at the probe, which leads to larger concentrations measured inside the bubble. Contrary to a gas-liquid bubbles, in a fluidized bed, the transition from bubble to suspension phase is not a sharp transition. Thus, a distribution of different low concentrations represents the bubble phase in the probability density distribution.

The dense phase concentration changes as a function of radial and axial measurement position and superficial gas velocity. Different locations of peaks have been found by evaluation of the different measurement points. Therefore, the same effects of concentration variation that happen in the bubble phase can occur also in this case.

This study was based on the two-phase theory of dense fluidized beds. Probability density distributions are used to define these phases, as developed by Werther and Molerus [3]. Peaks representing bubble and dense phase as shown in Figure 3 occur at each superficial gas velocity investigated. Peak concentrations of the dense phase decrease with increasing superficial gas velocity, meaning a larger expansion of the dense phase. By contrast, bubble phase peak concentrations are independent of the superficial gas velocity and close to a value of zero. The mean value of the peaks of dense phase and bubble phase results in the fitted Equation (3) as a definition of the phase border concentration ($c_{V,border}$) in dependence on the superficial gas velocity ($U_0$).

$$c_{V,border} = -0.017 \, U_0 + 0.242 \tag{3}$$

All concentrations above this phase border are assumed to belong to the dense phase. The repeated occurrence of concentrations below Equation (3) due to the penetration of bubbles by the probe was further analyzed to determine bubble properties.

The bubble phase holdup ($\phi_B$) is calculated according to Equation (4), which describes the probability densities ($P(c_V)$) of occurring bubble phase concentrations normalized by the integral of the whole probability density function.

$$\phi_B = \frac{\int_0^{c_{V,border}} P(c_V) \, dc_v}{\int_0^{c_{V,fb}} P(c_V) \, dc_v} \tag{4}$$

### 2.4. Determination of Bubble Properties

The determination of bubble properties, such as their velocity, frequency and length, follow an algorithm that detects the presence of a bubble. A minimum bubble length is defined for two reasons:

- Close passes of bubbles along the probe (probe not fully entering the bubble), as explained before (this is likely to happen in the turbulent state);
- Electrostatic charging in the fluidized bed (occurring at larger velocities and larger diameters with dry air), leading to spontaneous discharges visible in the probe voltage signal.

The minimum bubble length time ($t_{B,length,min}$) was set to 5 ms. For bubbles rising with a velocity of 2 m/s the minimum bubble size would therefore be registered by the algorithm as 1 cm, which is far lower than the size a bubble rising with this velocity should have according to the literature and measurements [1].

To prevent influences of electrostatic charging or short time crossing of the phase border on the bubble size, a minimum bubble distance time ($t_{B,distance,min}$) of 2 ms was introduced. This means that bubbles occurring within the minimum bubble distance time are counted as one and as separate bubbles otherwise. The minimum bubble distance time was chosen to be small because a larger time would cause bubbles following each other closely to be counted as one bubble. If bubbles rise with a velocity of 2 m/s in this case the bubbles can have a maximum distance of 4 mm to be counted as one bubble. This is unlikely to happen unless the bubbles already started coalescing.

Two channels are recorded simultaneously with one in the flow shadow of the other. Slightly different signals at both channels are the result. To be sure to determine bubbles, an overlap check of bubbles of both channels is done. All detected bubbles fulfilling the conditions of the algorithm are counted and give the bubble frequency $f_B$ when related to the signal length. For the determination of the bubble velocity ($U_B$), the time lag ($\tau$) between both channels is estimated. Equation (5) gives the normalized cross-covariance function ($\psi_{cV1,cV2}(\tau)$) that directly yields the Pearson correlation coefficient (R) of both signals for each time step of the signal shift.

$$\psi_{c_{V1}c_{V2}} = \lim_{T\to\infty} \frac{1}{2T\sigma_{c_{V1}}\sigma_{c_{V2}}} \int_{-T}^{T} (c_{V1}(t) - \overline{c_{V1}})(c_{V2}(t+\tau) - \overline{c_{V2}}) dt \tag{5}$$

where $\sigma_{cV1}$ and $\sigma_{cV2}$ are the standard deviations and $\overline{c_{V1}}$ and $\overline{c_{V2}}$ are the mean values of the solid concentrations $c_{V1}$ and $c_{V2}$ of the channels, respectively; $t$ is the time and $T$ is the maximum integration time.

This approach additionally offers a comparison of the signals according to their self-similarity. The correlation coefficients can vary between −1 and 1 with the highest value being the highest similarity in the signals. For the determination of the lag of both signals the time shift location of the peaks with the highest values is determined. Only signals with a correlation coefficient above 0.9 are considered to guarantee a high similarity in the signals and avoid errors in the measurement. The time lag of the peaks is then determined. The knowledge about the vertical distance ($d_{ch}$) of both channels to each other leads together with the time lag ($\tau$) to the velocity of the bubble ($U_B$) as given in Equation (6).

$$U_B = \frac{d_{ch}}{\tau} \tag{6}$$

The occurrence of negative bubble velocities was neglected in this study. Assuming a bubble is not accelerated while passing the capacitance probe, the duration a bubble needs to pass the probe ($t_B$) and its velocity give the pierced length ($l_p$) of a bubble in Equation (7).

$$l_p = U_B \, t_B \tag{7}$$

Coalescence and bubble formation are stochastic processes. Thus, bubble properties follow a natural distribution, which is a logarithmic normal distribution. The density function of the log-normal distribution $f(x)$ of the parameter $x$ is defined by Equation (8) [27].

$$f(x) = \frac{1}{\sqrt{2\pi}\sigma x} \exp\left(-\frac{(\ln(x) - \mu)^2}{2\sigma^2}\right) \tag{8}$$

where $\mu$ and $\sigma$ are the expected value and the standard deviation of the parameter's natural logarithm, respectively. The arithmetic mean value ($E(X)$) is given by Equation (9):

$$E(X) = \exp\left(\mu + \frac{1}{2}\sigma^2\right), \tag{9}$$

and the arithmetic standard deviation ($SD(X)$) is given by Equation (10).

$$SD(X) = \exp\left(\mu + \frac{1}{2}\sigma^2\right)\sqrt{\exp(\sigma^2) - 1} \tag{10}$$

Examples for histograms of the pierced lengths measured at two different superficial gas velocities are given in Figure 4. The log-normal distributions fit the measured histograms well. Higher gas velocities show a broader distribution of pierced lengths.

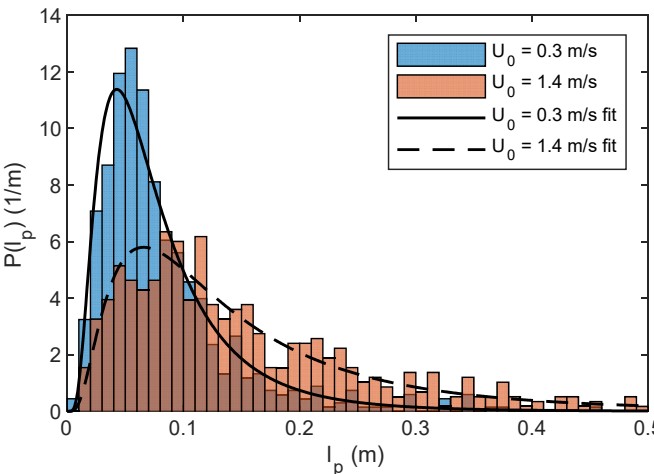

**Figure 4.** Probability density histogram of the pierced lengths measured at the center (r/R = 0) of the fluidized bed at $D = 0.1$ m, $H_0 = 0.3$ m and $z = 0.1$ m at bubbling fluidization ($U_0 = 0.3$ m/s) and turbulent fluidization ($U_0 = 1.4$ m/s) and their fitted log-normal distributions.

The same behavior as for the pierced lengths can be found for the bubble velocities in Figure 5. Lags in the histogram at large velocities can be explained by the measurement frequency and the resulting minimum recorded time step, which has a large influence as given by Equation (6). Furthermore, the deviation of the histograms from the fitted log-normal distributions is larger than for the pierced lengths. Reason for these deviations are velocity variations due to mutual interactions of the bubbles (e.g., acceleration due to coalescence). These interactions increase with larger amounts of bubbles resulting in the largest fitting errors occurring in the turbulent regime.

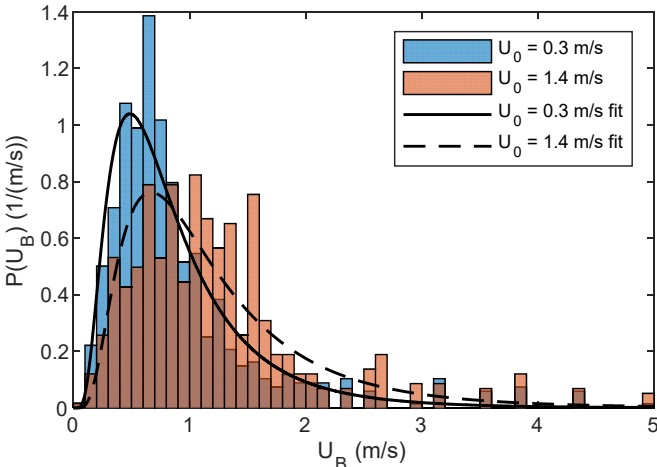

**Figure 5.** Probability density histogram of the bubble velocity measured at the center (r/R = 0) of the fluidized bed at $D = 0.1$ m, $H_0 = 0.3$ m and $z = 0.1$ m at bubbling fluidization ($U_0 = 0.3$ m/s) and turbulent fluidization ($U_0 = 1.4$ m/s) and their fitted log-normal distributions.

The arithmetic mean values of the distribution functions calculated by Equation (9) were used in this study to compare the bubble velocities and mean pierced lengths under different conditions. Furthermore, the mean pierced length and the mean bubble velocity were cross-sectional averaged, separately. This was done in relation to the cross-sectional area belonging to a radial measurement position and the amount of bubbles occurring in this area.

## 3. Results and Discussion

### 3.1. Bubble Phase Holdup

As expected, the bubble phase holdup increased with increasing superficial gas velocity. The intensity of the change of the bubble phase holdup depended on the fluidized bed diameter, as can be seen in Figure 6. At larger diameters, the holdup of the bubble phase was generally smaller than in small plants. The main reason for this observation is the much larger sizes that bubbles can grow to in a larger bed and consequently pass through the bed with higher velocity, as bubble velocity is a function of bubble size. This relationship is expected for small laboratory-scale units, where bed diameter is in the order of magnitude of bubble size. Therefore, no impact is expected above certain bed diameters and bed heights. In large fluidized beds, bubbles and the downflow of solids develop preferred stable flow patterns [4], where the interaction between up- and downflow is lower than in small beds. The resulting smaller bubble holdup in larger beds also leads to a smaller bed expansion. These impacts are expected both for bubbling (mainly for group B particles) and turbulent fluidized beds.

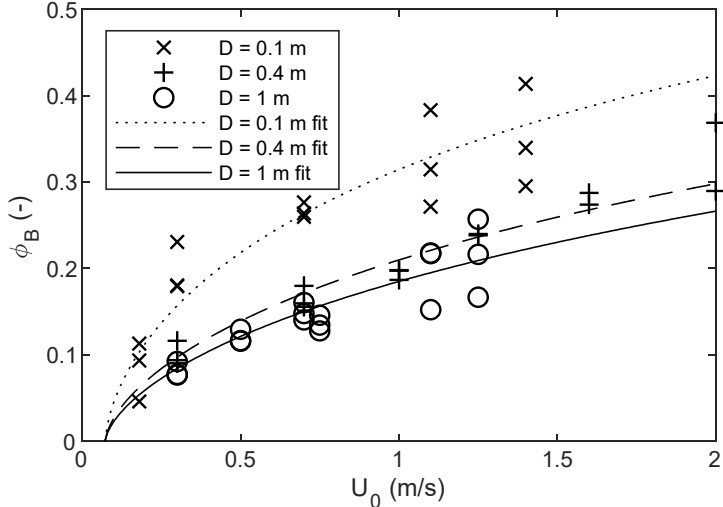

**Figure 6.** Cross-sectional averaged bubble holdups in dependence of superficial gas velocity and bed diameter at different heights above the gas distributor.

Fluctuations of bubble phase holdup showed no clear correlation with the measurement height. In the fluidized bed with a diameter of $D = 0.4$ m and a static bed height of $H_0 = 0.8$ m, the phase holdup was measured at six positions above the gas distributor. The measurement position showed no significant influence on the cross-sectional averaged bubble holdup. Close to the bed surface, the bubble phase holdup decreased. This phenomenon occurs due to the eruption of bubbles.

A correlation was developed to model bubble holdup ($\phi_B$) as a function of bed diameter (D), superficial gas velocity ($U_0$) and the minimum fluidization velocity ($U_{mf}$) based on the experimental results:

$$\phi_B = \frac{1}{1 + \left( \frac{4.735D}{0.128+D} \right) \left( U_0 - U_{mf} \right)^{-0.641}} \tag{11}$$

In this correlation, it is assumed that the bubble holdup is independent of the height above the gas distributor, as discussed above.

The correlation fits the experimental data well (Figure 6), considering the large spread in the experimental data for small diameters. The parity plot (Figure 7) confirms reasonable agreement for larger fluidized beds, with most predictions being within a band of ±10% deviation as shown in the

parity graph (Figure 7). The largest errors occurred for the smallest diameter of 0.1 m. The main reason is the large influence of bubble size in small diameters as mentioned above. With increasing height above the gas distributor, larger bubbles are formed due to coalescence [3,6,13,24,28]. These bubbles interact with the downflowing solids at the wall, which leads to a higher bubble phase holdup. Thus, an influence of the measurement height above the gas distributor can occur in small beds. Radial profiles of the bubble phase hold up in this study show that small amounts of bubbles can move in the wall region in small beds. By contrast, almost no bubbles occurred in the wall region in large beds, which proves a larger interaction of rising bubbles and downflowing solids in small beds. The wall effect reduced as bed diameter increased as can be observed in the results, where the difference between 0.4 m bed and 1 m bed results was much smaller than between the 0.1 m bed and 0.4 m bed. The difference became less significant for larger bed diameters.

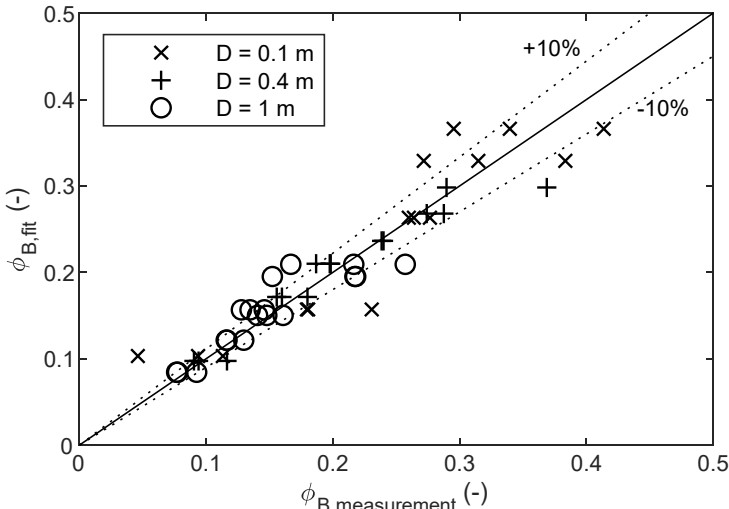

**Figure 7.** Parity plot of the measured bubble phase holdup and the developed correlation for all investigated plant diameters.

Equation (11) was developed for only one fraction of bed material classified to Geldart group B. Influences of particle properties are considered in the minimum fluidization velocity but the applicability for other beds needs further investigation of the model parameters with regard to particle properties.

*3.2. Bubble Size*

The distribution of pierced lengths of bubbles and the resulting arithmetic mean value depend on the radial position and axial position of the measurement as well as on superficial gas velocity and bed properties. Larger mean pierced lengths were found to occur in regions of deprived bubble flow due to coalescence behavior. These phenomena are well described by Werther and Molerus [4].

The cross-sectional averaged arithmetic means of the pierced length ($l_p$) as a function of superficial gas velocity ($U_0$) for different heights above the gas distributor ($z$) are plotted in Figures 8 and 9. The tests were carried out on the plants with diameters of 0.1 and 1 m. The mean pierced length increased with higher superficial gas velocities as with the bubble phase holdup. Furthermore, a clear dependence of the mean pierced length on the height above the gas distributor could be found. This behavior was expected due to coalescence phenomena occurring more likely at higher superficial gas velocities and resulting in larger bubbles above the gas distributor. In the literature, the transition from bubbling to turbulent fluidization is assumed to involve a decrease of average bubble sizes, which lead to a decrease of pressure fluctuations in the turbulent regime [16]. In this study, bubbles with larger mean pierced lengths were also found in the well-developed turbulent regime. The experimental

data of the pierced bubble length from three plants were used to develop a correlation expressed in Equation (12):

$$l_p = 0.751 \, z^{0.356 \frac{(D+0.089)}{D}} \left(U_0 - U_{mf}\right)^{0.148 \frac{(D+0.232)}{D}} \tag{12}$$

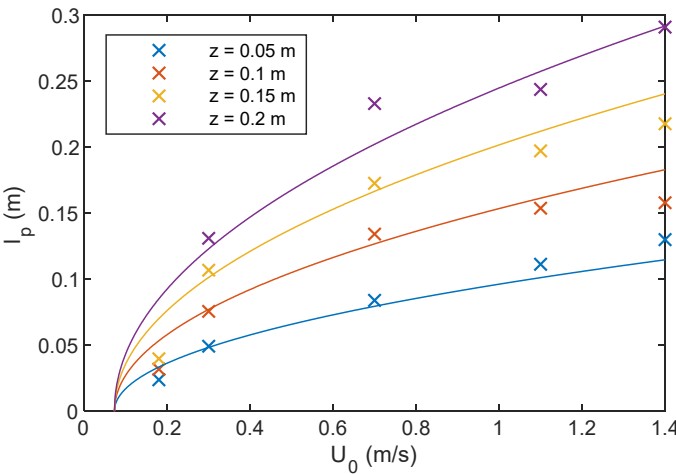

**Figure 8.** Cross-sectional averaged mean pierced lengths measured at different heights above the gas distributor compared with the correlation prediction as function of superficial gas velocity in the FB100 unit.

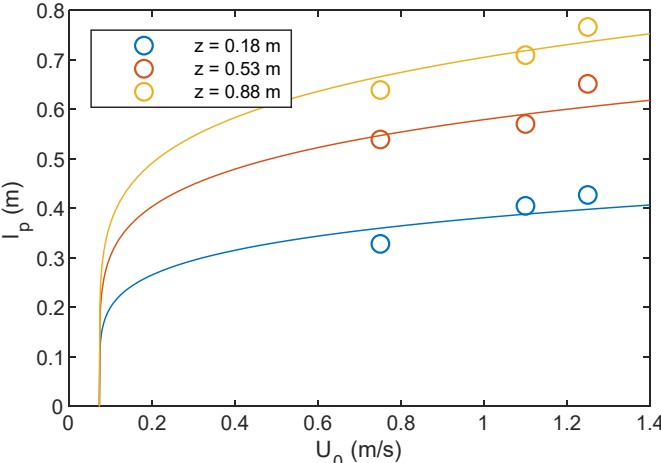

**Figure 9.** Cross-sectional averaged mean pierced lengths measured at different heights above the gas distributor compared with the correlation prediction as function of superficial gas velocity in the FB1000 unit.

Figures 8 and 9 show correlation results compared to the experimental data. The correlation prediction was in good agreement with the measured cross-sectional averaged mean pierced lengths, especially at larger velocities. The correlation overestimated the mean pierced length at low gas velocities. The mean pierced lengths were small compared to the bed diameter of 0.1 m at this velocity. This difference is mainly related to the wall effect, where the shape of bubbles is impacted mainly in smaller beds particularly at higher elevations above the distributor at higher gas velocities. Bubbles of this size (e.g., $l_p = 0.13$ m at $z = 0.2$ m and $U_0 = 0.3$ m/s) are clearly influenced by the size of the fluidized bed, resulting in vertically stretched bubbles. In addition, bubbles are compressed by the backflowing solids along the wall of the plant, resulting in larger mean pierced lengths in comparison to smaller bubbles at a superficial gas velocity of 0.18 m/s. This effect is partially covered by the bed diameter

terms in the correlation. This dependence on the bed diameter in the correlation is mostly present at smaller diameters. This can be observed in Figure 9, where the correlation is in good agreement with the experimental results while the bed diameter term has a negligible effect. The correlation helps in extrapolating small-scale lab unit results to larger scales.

Comparison of the measured pierced lengths at heights of $z = 0.2$ m in Figure 8 and $z = 0.18$ m in Figure 9 with similar operating conditions for two different bed diameters showed that bubbles had larger mean pierced lengths at larger beds under similar conditions. One reason for this observation is the higher probability of coalescence in a larger bed. Coalescence can take place horizontally and vertically whereas vertical coalescence is more likely according to the literature [29–33]. However, these observations are probably influenced by bed diameter. If a bubble in a small plant reaches a size close to the diameter of the fluidized bed, lateral coalescence cannot take place because it is the only bubble rising in the cross-sectional area. It is unlikely that following bubbles have a larger velocity than the leading one, which reduces probability of vertical coalescence in this case. Therefore, a semi-slugging state will be reached as bubble diameters cannot reach the full diameter of the fluidized bed. In a larger fluidized bed, lateral and vertical coalescence can still take place, supporting higher probability for the formation of larger bubbles. Another reason for larger bubbles in larger fluidized beds is the formation of larger bubbles at the gas distributor [1]. Here, regions of deprived bubble flow and solids backflow can be larger compared to a small fluidized bed. Because weaker shear forces occur in large fluidized beds, the formation of large bubbles in the regions of deprived bubble flow is not hindered. This increases the probability of forming larger bubbles. The parity plot in Figure 10 shows that Equation (12) is in good agreement with experimental results. The largest errors occur at low velocities in the bubbling regime as discussed before.

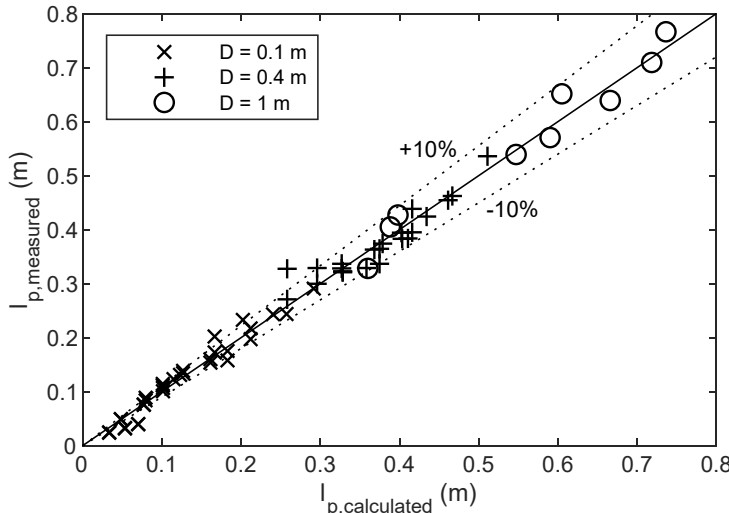

**Figure 10.** Parity plot of correlation prediction vs. experimental result of cross-sectional averaged mean pierced length for all plant diameters investigated.

Geldart group B particles do not have a distinct bubble size limit, in contrast to group A particles [15]. The impact of particle density and diameter was taken into account via the minimum fluidization velocity ($U_{mf}$) in the developed correlations. However, further experimental tests are required regarding this issue. In this study, only the vertical dimension of bubbles was investigated. As described in the introduction, capacitance probe measurements of bubbles do not yield any information about horizontal bubble properties without assumptions about the bubble shape. Bubbles in turbulent fluidized beds do not have a regular shape as in bubbling fluidized beds. They have a much more irregular and dispersed form, as shown by Holland et al. [23]. Visual observations in this study confirmed the irregular shape of bubbles. The distinction between both phases under assumption of the two-phase theory was difficult visually, and the gas seemed to rise in vertical stretched gas

pockets or voids with small horizontal sizes. These observations agree with the vertical sizes of bubbles measured in this study. The bubbles in all plants reached vertical sizes in the range of or larger than the diameter of the fluidized bed itself. Bubbles having a volume-equivalent diameter of this size would lead to slugging, which cannot be observed. This leads to the conclusion that bubbles in the turbulent regime must be much smaller in the horizontal dimension than in the vertical dimension, particularly in small beds. The transition of the shape from spherical caps in the bubbling regime to the vertical stretched shape at turbulent fluidization is a continuous process as the measurements showed. The vertical length of bubbles rose monotonously with increasing superficial gas velocity.

This subject needs further investigations to determine the shape of bubbles and particularly the impact of the bed on their horizontal dimension in order to be able to use this information for the scale-up of processes.

### 3.3. Bubble Velocity

Most studies in the literature about the velocity of bubbles concentrated on superficial gas velocities of up to $U_0 = 0.3$ m/s [1]. This study also investigated bubble velocities up to superficial gas velocities of $U_0 = 1.6$ m/s, covering the upper region of the bubbling regime up to turbulent fluidization. Figure 11 shows all measurements of the cross-sectional averaged mean bubble velocities as a function of their mean pierced length. In contrast to the relationship of volume-equivalent bubble diameter to bubble velocity of ($U_B \sim d_{B,V}^{0.5}$) found in the literature [1], a linear relationship between pierced length $l_p$ and bubble velocity $U_B$ was found. This relationship is given by Equation (13):

$$U_B = 0.587 \frac{U_0 - U_{mf}}{U_0} + 4.344 \, l_p \tag{13}$$

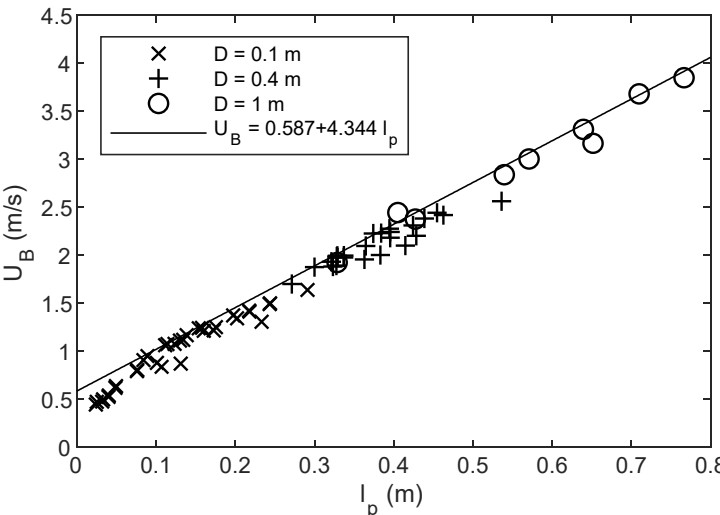

**Figure 11.** Bubble velocity as a function of the mean pierced length at different plant diameters, heights above the gas distributor and superficial gas velocities (cross-sectional averaged) including Equation (13) at $U_0 \gg U_{mf}$ as a limiting case.

Correlations developed only for the bubbling regime result in an overestimation of the velocity of bubbles, which was also found in the literature [16,22]. Equation (13) was developed under the condition that the superficial gas velocity be much higher than the minimum fluidization velocity ($U_0 \gg U_{mf}$). Superficial gas velocities above 1 m/s almost fulfill this condition and the relation of bubble velocity to pierced length is similar to the solid line plotted in Figure 11. At lower superficial gas velocities (e.g., $U_0 = 0.18$ m/s), the ratio of both parameters depends on the superficial gas velocity. In bubbling fluidization such a behavior was also observed in the literature and explained by the swarm

behavior of bubbles [24,34]. The rise of bubbles in a swarm and in the wind shadow leads to larger bubble rise velocities. At higher superficial gas velocities the influence of swarm behavior is limited. This behavior can be observed in the experimental results from this study and developed equation.

Equation (13) fits the experimental data with a maximum error of 12%, as can be seen in Figure 12. The largest errors occur at large measurement heights above the gas distributor in comparison to the diameter of the fluidized bed (in most cases z/D > 1.5) in the bubbling fluidized bed regime. In these cases, the bubbles were large enough in horizontal size to be influenced by the bed diameter. The increased resistance due to solids moving downward at the wall increased the drag force acting on the bubbles, which slowed them down and deformed the bubbles.

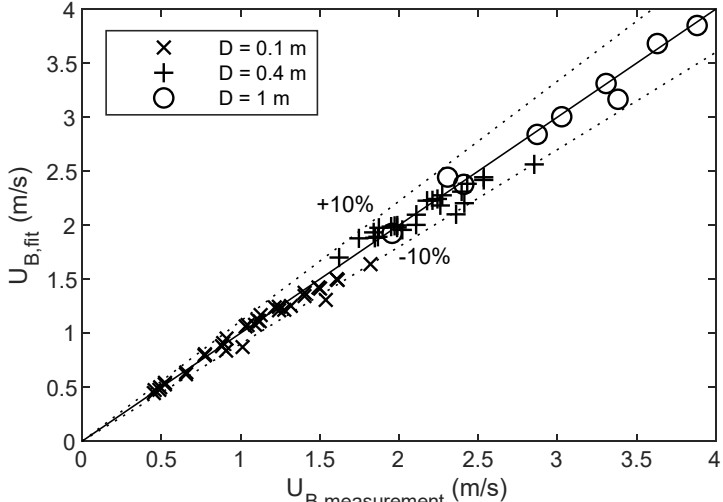

**Figure 12.** Parity plot of measured bubble velocities and fitted velocities as a function of the mean pierced length of the bubbles.

The formation of bubbles in a fluidized bed happens due to the tendency of the system to minimize energies. Due to the fact that bubbles in fluidized beds have no surface tension [2,24], the shape of a bubble is formed to minimize the drag force acting on the bubble. Bubbles in (bubbling) fluidized beds behave like bubbles in highly viscous liquids [30]. At low velocities, they were found to have a shape of a spherical cap with a drag coefficient ($C_D$) of 2.64 [30]. By contrast, the measurements in the turbulent fluidized bed regime show a change of the bubble shape, which must lead to a change of the drag coefficient of a bubble. The drag coefficient of a bubble can be determined via a force balance of drag force ($F_D$), buoyancy force ($F_B$) and gravity force ($F_G$), as given by Equation (14).

$$F_D = F_B - F_G \tag{14}$$

The drag force is defined by Equation (15) with the drag coefficient $C_D$, the density of the fluidized bed $\rho_{fb}$, the horizontal cross-section of the bubble $A_B$ and the bubble velocity $U_B$:

$$F_D = \frac{1}{2}C_D\, \rho_{fb}\, A_B\, U_B{}^2, \tag{15}$$

the buoyancy force is defined by Equation (16) with the density of the fluidized bed $\rho_{fb}$, the volume of the bubble $V_B$ and the gravitational acceleration g:

$$F_B = \rho_{fb}\, V_B\, g \tag{16}$$

and the gravity force is defined by Equation (17) with the density of the fluid $\rho_f$, the volume of the bubble $V_B$ and the gravitational acceleration $g$:

$$F_G = \rho_f \, V_B \, g. \tag{17}$$

Under the assumption that bubbles are ellipsoids having a horizontal diameter $b$ and a vertical diameter equal to the pierced length $l_p$, the volume $V_B$ and the bubble cross-sectional area $A_B$ are defined as in Equation (18).

$$V_B = \frac{\pi}{6}b^2 \, l_p \; ; \; A_B = \frac{\pi}{4}b^2 \tag{18}$$

The density of a fluidized bed is more than two order of magnitudes higher than the gas density at ambient conditions ($\rho_{fb} >> \rho_f$), which leads to the conclusion that gravity force acting on a bubble is negligible in comparison to the buoyancy force. This leads to the drag coefficient definition in Equation (19).

$$C_D = \frac{4}{3}g\frac{l_p}{U_B{}^2} \tag{19}$$

The calculated drag coefficients (from measured cross-sectional averaged mean pierced lengths and bubble velocities) are shown in Figure 13 together with curves of the drag coefficient determined by combining Equations (19) and (13) at different superficial gas velocities. The calculated drag coefficients can be smaller than the ones in reality because the swarm effect is not considered in this calculation. This can be a reason for calculated drag coefficients ($C_D$) lower than a value proposed by the literature of 2.64 in the bubbling regime [30].

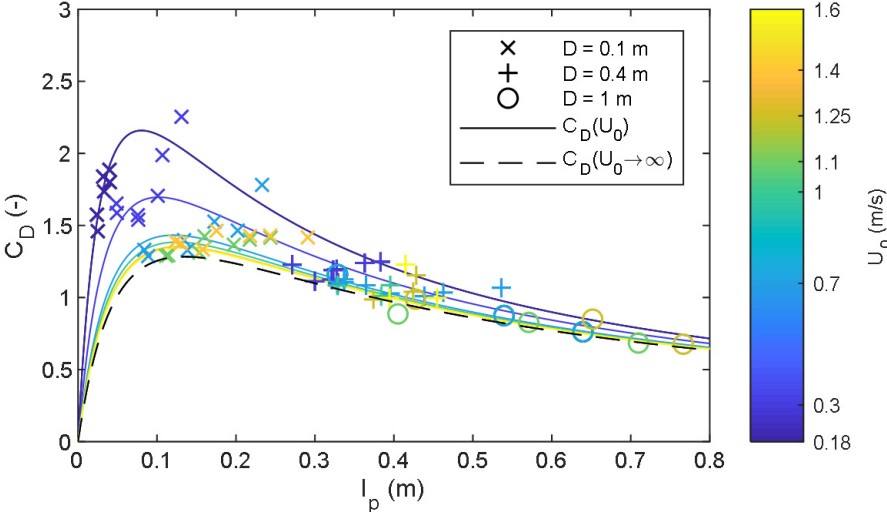

**Figure 13.** Bubble drag coefficient as a function of the bubble mean pierced length.

According to Figure 13, the drag coefficient first increases as long as the pierced length is below the range of 0.08–0.13 m (depending on superficial gas velocity); it then reaches a maximum and decreases when they grow further. Bubbles in the first (increasing) section occur in the bubbling fluidized regime. These bubbles contain a small volume of gas, which precludes the formation of a large wake, especially given the high inertia of Geldart group B particles. Therefore, these bubbles might have a shape closer to a sphere. With growth of bubble size, the formation of a larger wake is probable. These large bubbles must carry more material in their wake in comparison to their size compared to smaller bubbles [35]. This leads to a change in shape tending to the bubble cap form the one often described in the literature [1]. If bubbles grow further, the inertia of the solids and the large forces acting on the bubble result in higher deformation of the bubble. The resulting shape is a vertical stretched one and increasing pierced lengths can be measured. Furthermore, a smaller horizontal

bubble diameter does not allow the formation of an even larger wake. Thus, the change of the shape results in a lower drag coefficient for larger bubble sizes even in the bubbling fluidized bed regime.

With increasing superficial gas velocity, the bubble velocities increase as in Equation (13). The increasing bubble velocities lead to larger forces acting on the bubbles, which in turn change the shape of the bubble and wake. A reduction of the drag coefficient at larger superficial gas velocities is the result that can be seen in Figure 13.

### 3.4. Effect of Bubble Size and Shape on the Transition from Bubbling to Turbulent Fluidization

The transition from bubbling to turbulent fluidization is defined by a maximum in the pressure fluctuations [16]. Pressure fluctuations occur due to bubble formation, coalescence, bubble break-up and bubble eruption at the surface. Large bubbles occurring in a fluidized bed induce high pressure fluctuations due to their large pressure gradients at cloud and wake and the displacement of large amounts of solids. The displacement of solids and the forces acting on a bubble increase with its cross-sectional area. It was found in this study that bubbles must change their horizontal size when they grow due to an increase of the superficial gas velocity, while the amount of displaced solid and the forces acting per bubble volume decreases as a result. At the same time, the amount of bubbles rising increases with increasing superficial gas velocity, as measurements of the local bubble frequency showed. Due to an increase in the coalescence rate and the formation of larger bubbles at the gas distributor at high superficial gas velocities, the increase of the bubble frequency stagnates. At a certain superficial gas velocity, the reduction of the bubble volume-dependent forces overcome the increasing number of bubbles. Pressure fluctuations reach a maximum at this point and decrease with a further increase of the superficial gas velocity.

Due to the wall effects explained in Sections 3.1 and 3.2 and the tendency of the system to reduce drag force, bubbles in small fluidized beds are forced to coalesce even at larger superficial gas velocities. This leads to a later transition into the turbulent regime at lower bed diameters at the same static bed heights, which was reported in the literature [16,26,36–38]. The size of bubbles increases with distance from the gas distributor due to coalescence. These larger bubbles must change their shape at lower superficial gas velocities in contrast to smaller bubbles at the bottom of the fluidized bed in order to minimize energy. This explains why the turbulent bed is normally formed in the top section of the bed and developed downwards into the bottom section at larger superficial gas velocities, as observed in the literature [39,40].

## 4. Conclusions

Capacitance probes were used for the measurement of vertical bubble sizes and bubble velocities. Bubble phase behavior was investigated over a large range of superficial gas velocities ranging from bubbling fluidization ($U_0$ of 0.18 m/s) up to turbulent fluidization ($U_0$ of 1.6 m/s). Experiments were carried out in laboratory scale-fluidized bed facilities of different sizes of 0.1, 0.4 and 1 m in diameter.

Results of the investigation showed a clear dependence of the bubble phase holdup on the fluidized bed diameter and the superficial gas velocity. Bubble holdup was higher in beds with a smaller diameter. This was mainly due to the wall effects on the bubbles, which attain sizes comparable with the bed diameter.

An empirical correlation was developed for the prediction of the cross-sectional averaged mean pierced length as a function of bed diameter, height above the gas distributor and superficial gas velocity. The vertical mean pierced length was found to grow even in the turbulent fluidization regime. Large measured mean pierced lengths in the turbulent regime led to the conclusion that the shape of bubbles was influenced by wall effects, superficial gas velocity and the size of the bubble itself. Small bubbles at low superficial gas velocity had a higher probability to have a spherical shape and spherical cap, whereas larger bubbles and velocities led to a deformation of the bubble into a vertical stretched shape.

The velocity of a bubble mainly depends on its vertical size. Experimental results showed a linear dependence of the bubble velocity on its pierced length. The influence of the superficial gas velocity and thus the influence of swarm behavior are limited. The transition from bubbling to turbulent fluidization was found to mainly depend on the shape of the bubbles. The occurrence of vertical stretched bubbles having a small horizontal size compared to their vertical dimension leads to a reduction in the pressure fluctuations.

**Author Contributions:** Conceptualization, T.W.; methodology, T.W.; validation, T.W.; formal analysis, T.W.; investigation, T.W.; resources, S.H.; data curation, T.W.; writing—original draft preparation, T.W.; writing—review and editing, M.Y. and S.H.; visualization, T.W.; supervision, M.Y. and S.H.; project administration, S.H. All authors have read and agreed to the published version of the manuscript.

**Funding:** This research was funded by Total Research and Technology Gonfreville (TRTG), 76700 Harfleur, France.

**Acknowledgments:** In memory of Ernst-Ulrich Hartge, project supervisor and expert in this field. Technical support, production of capacitance probes and facility maintenance were done by Heiko Rohde at the Institute of Solids Process Engineering and Particle Technology, Hamburg University of Technology. Language revision by Paul Kieckhefen.

**Conflicts of Interest:** The authors declare no conflict of interest.

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
