# Peer review of "Bubble Properties in Bubbling and Turbulent Fluidized Beds for Particles of Geldart’s Group B"

_processes, doi:10.3390/pr8091098_

Round 1
Reviewer 1 Report
The present work studies bubble properties for Geldart B particles for different fluidization regimes. The manuscript is well-written and structured and will significantly contribute to that scientific field. A highlight is the use of three fluidized-bed test rigs with different sizes, which allows a comprehensive analysis of bubble properties for a wide range of parameters. I recommend the acceptance of the manuscript for publication.
Some minor comments that should be considered:
Introduction: In the last paragraph it could be mentioned that the focus of the paper is not only to study higher velocities than existing literature, but also to analyze bubble properties such as shape and size in more detail.
Section 3.3: Please provide the bubble Reynolds numbers together with the drag coefficients, which may help to interpret the data by comparing against existing drag correlations from literature.
Line 484: Due to the wall effects …
Author Response
Dear reviewer,
Please find my answers to your review in the attachment.

Reviewer 2 Report
This study measures the gas holdup, bubble velocity and bubble size for three fluidized bed setups using capacitance probes and develops correlations for these quantities. The data is interesting, since it covers a large range of setups and conditions and is therefore useful for understanding fluidized bed behaviour and potentially for the validation of numerical simulations. However, several factors detract from the quality of the paper and large modifications will be required to reach a sufficient standard for publication. The reviewer therefore recommends re-evaluation after a major revision.
Major changes:
- In general the quality of writing is of insufficient quality and present serious challenges to following the arguments of the authors, thereby detracting from the research. The reviewer has pointed out some issues (included a list of minor language issues in the attached PDF), but a major overhaul of the paper is required.
- The novelty of the contributions in the paper is unclear. A vague statement is made about insufficient data in the turbulent regime, but introduction should be expanded to clearly show how this paper closes the knowledge gaps. Also, the authors should address more clearly how their correlations improve on the many existing closures in literature, for example by comparing their experimental data against existing closures.
Compulsory changes:
- Introduction: Although the authors argue that not many studies have considered the turbulent regime, such studies certainly exist. The introduction should include a review of the findings from the studies that did look at bubble properties in the turbulent regime and a discussion of the shortcoming of these studies that justifies the existing study.
- Line 192: The two-phase theory of dense fluidized beds is utilized to determine the bubble properties. However, the primary justification for this paper is to study the turbulent regime, where the bed is less dense. The author should therefore discuss the validity of this assumption, and the correlation used in equation 3, when investigating the turbulent regime.
- Line 216: The description should be improved to clarify that this is referring the minimum time between bubbles for them to be counted as separate bubbles.
- Figure 5: The log-normal distribution does not fit the data very well. The authors should comment on this.
- Line 267: The authors should comment on the variation of the mean pierced length with radial position and the effect of the cross-sectional average that is employed.
- Equation 11: Since all cases here use the same particles, the authors should comment on the validity of their correlation for fluidized beds with different minimum fluidization velocities.
- Line 298: There must be an error in this claim, since the data in Figure 6 shows that some the correlation is more than 10% away from some experimental points for the smallest bed. Also, the values in Figure 6 (0-0.45) does not match the values in the parity plot in Figure 7 (0.55-1) (which appears to be for the solids holdup) therefore there appears to be an error in this analysis.
- Line 309 – From where is the value of exactly 0.6 taken? The data in this study is not sufficient to claim this exact number, if it is from literature it should be cited.
- Line 337 – The authors should include lines in Figure 9 where D = 0 to demonstrate this claim.
- Figure 13 should be modified to allow a direct comparison between the experimental and correlation data, e.g. by having a colour scale to show the superficial gas velocity for each experimental point.
Minor recommended changes:
- Abstract: The argument for the last two sentences are unclear, therefore these sentences should be revised.
- Abstract: It would be beneficial to add more quantitative results as part of the abstract.
- Line 79: Some references should be added for the studies that have focussed on the bubbling fluidized bed regime.
- Line 107: It would be good to also mention the particle properties for which these transition velocities were determined.
- Line 161: This sentence is confusing. Is this saying that the measurement is at the wall? And it is assumed "plant" should be bed.
- Line 209: This explanation should be clarified.
- Line 212: Referring to a length, but giving a unit of time.
- Line 295: In would be good to refer back to the earlier comment that this assumption is supported by the experimental data.
- Figure 11: In the caption, refer to the equation in which the correlation is given.
- Line 304: Is this data from this study or referring to literature?
- In general it would be beneficial to better show in the figures plotted against superficial velocity, where the transition from bubbling to turbulent conditions is expected to occur.

Author Response

(The authors gave the same response as above.)

Reviewer 3 Report
Comments on the article “Bubble properties in bubbling and turbulent fluidized beds for
particles of Geldart’s group B” submitted to Processes.
Concluded Remark: Major Revision
The paper deals with a thorough investigation about the size, 13 shape and velocity of bubbles was done at superficial gas velocities. The paper can be interesting for readers provided some necessary modifications are done. The article involves many examples of language errors. Some suggestions and comments related to the articles must be addressed which are given below. After considering the following suggestions, the article can be taken into consideration for publications.
1. The introduction seems to be very concise and limited to only a particular area. For broadening the audiences, the introduction should contain some references related to sprayed fluidized bed granulator and twin-screw granulator. Some suggestions related to the references are given below:
(a) M. Hussain, J. Kumar, E. Tsotsas (2016), Micro-macro transition of population balances in fluidized bed granulation, Procedia Engineering, 102,1399-1207.
(b) G Kaur, M Singh, T Matsoukas, J Kumar, DB Thomas, I Nopens (2018) Two-Compartment Modeling and Dynamics of Top-Sprayed Fluidized Bed Granulator, Applied Mathematical Modelling, Vol. 68, 267--280.
(c) Liu H., Li M. (2014) Two-compartmental population balance modeling of a pulsed spray fluidized bed granulation based on computational fluid dynamics CFD analysis. J. Pharm., 475 (1), 256-269.
(d) G Kaur, M Singh, J Kumar, T De Beer, I Nopens (2018) Mathematical modeling and simulation of sprayed fluidized bed granulator, Processes, Vol. 6(10), 195.
2. I would urge the authors to describe the difference between your current work over the recent article [18] published in Powder Technology.
3. There are many language errors. I believe the author must read the manuscript carefully and improve the language rigorously.
Author Response

(The authors gave the same response as above.)

Round 2
Reviewer 3 Report
Dear Editor,
The authors have made significant changes in the manuscript. Accept in the present form.